# Cerebral [^18^F]-FDOPA Uptake in Autism Spectrum Disorder and Its Association with Autistic Traits

**DOI:** 10.3390/diagnostics11122404

**Published:** 2021-12-20

**Authors:** Rik Schalbroeck, Lioe-Fee de Geus-Oei, Jean-Paul Selten, Maqsood Yaqub, Anouk Schrantee, Therese van Amelsvoort, Jan Booij, Floris H. P. van Velden

**Affiliations:** 1School for Mental Health and Neuroscience, Maastricht University, 6229 ER Maastricht, The Netherlands; j.selten@rivierduinen.nl (J.-P.S.); t.vanamelsvoort@maastrichtuniversity.nl (T.v.A.); 2Rivierduinen Institute for Mental Healthcare, 2333 ZZ Leiden, The Netherlands; 3Section of Nuclear Medicine, Department of Radiology, Leiden University Medical Center, 2333 ZA Leiden, The Netherlands; l.f.de_geus-oei@lumc.nl (L.-F.d.G.-O.); f.h.p.van_velden@lumc.nl (F.H.P.v.V.); 4Biomedical Photonic Imaging Group, University of Twente, 7522 NB Enschede, The Netherlands; 5Department of Radiology and Nuclear Medicine, Amsterdam University Medical Centers, Location VU Medical Center, 1081 HV Amsterdam, The Netherlands; maqsood.yaqub@amsterdamumc.nl; 6Department of Radiology and Nuclear Medicine, Amsterdam University Medical Centers, Location Academic Medical Center, 1105 AZ Amsterdam, The Netherlands; a.g.schrantee@amsterdamumc.nl (A.S.); j.booij@amsterdamumc.nl (J.B.)

**Keywords:** autism spectrum disorder, autistic traits, [^18^F]-FDOPA, positron emission tomography, dopamine, monoamine

## Abstract

Dopaminergic signaling is believed to be related to autistic traits. We conducted an exploratory 3,4-dihydroxy-6-[^18^F]-fluoro-L-phenylalanine positron emission tomography/computed tomography ([^18^F]-FDOPA PET/CT) study, to examine cerebral [^18^F]-FDOPA influx constant (*k*_i_^cer^ min^−1^), reflecting predominantly striatal dopamine synthesis capacity and a mixed monoaminergic innervation in extrastriatal neurons, in 44 adults diagnosed with autism spectrum disorder (ASD) and 22 controls, aged 18 to 30 years. Autistic traits were assessed with the Autism Spectrum Quotient (AQ). Region-of-interest and voxel-based analyses showed no statistically significant differences in *k*_i_^cer^ between autistic adults and controls. In autistic adults, striatal *k*_i_^cer^ was significantly, negatively associated with AQ attention to detail subscale scores, although Bayesian analyses did not support this finding. In conclusion, among autistic adults, specific autistic traits can be associated with reduced striatal dopamine synthesis capacity. However, replication of this finding is necessary.

## 1. Introduction

The atypical functioning of dopaminergic and other monoaminergic systems has long been hypothesized to contribute to autistic traits [1]. For instance, according to the dopamine hypothesis of autism spectrum disorder (ASD; [2,3]), alterations in the midbrain dopaminergic system are associated with clinical and sub-clinical autistic traits, including difficulties in social interaction and communication, and stereotyped behaviors. However, there is a lack of studies assessing in vivo monoamine functioning in autistic adults [4].

Positron emission tomography/computed tomography (PET/CT) studies have used 3,4-dihydroxy-6-[^18^F]-fluoro-L-phenylalanine ([^18^F]-FDOPA) to assess the presynaptic dopamine synthesis capacity in ASD. One study reported that [^18^F]-FDOPA uptake was decreased in the anterior medial prefrontal cortex in (mostly sedated) autistic children (*n* = 14), relative to typically developing controls (*n* = 10) [5]. Another study found that [^18^F]-FDOPA uptake was increased in the frontal and striatal regions in adults with Asperger syndrome (*n* = 8) relative to controls (*n* = 5) [6]. However, in both studies, sample sizes were small, and the associations with measures of autistic traits were not examined.

The social defeat hypothesis of schizophrenia posits that a subordinate or outsider position leads to an increased baseline activity or sensitization of the mesolimbic dopamine system and, thereby, to an increased risk of schizophrenia [7]. Since ASD is a risk factor for schizophrenia [8], we recently conducted a large [^18^F]-FDOPA PET/CT study to test the pre-registered hypothesis of increased striatal dopamine synthesis capacity in non-psychotic individuals with ASD [9]. Contrary to our hypothesis, the results indicated no differences in striatal [^18^F]-FDOPA uptake between individuals with ASD (*n* = 44) and controls (*n* = 22), and no association between this uptake and social defeat.

Here, we extend our previous study with exploratory region of interest (ROI) and voxel-based analyses, in which we compare striatal as well as extrastriatal [^18^F]-FDOPA uptake between adults with ASD and controls, and examine their associations with self-reported autistic traits.

## 2. Materials and Methods

### 2.1. Participants and Procedures

The full procedures are described in our previous publication [9]. We recruited Dutch participants aged 18 to 30 years, who were abstinent from current or recent psychotropic medication use (see Appendix A for details). Those with ASD, had received their diagnosis from a registered mental health clinician, and this diagnosis was confirmed by the first author using the Autism Diagnostic Observation Schedule-2 (ADOS-2) module 4 [10,11]. We included 44 autistic participants and 22 controls (frequency-matched on age, sex, and smoking status). All the participants provided informed consent. The study was approved by the medical ethics committee of the Leiden University Medical Center (reference NL54244.058.15).

### 2.2. Autism Spectrum Quotient

The Autism Spectrum Quotient (AQ) is a 50 item self-report questionnaire that assesses the presence of autistic traits [12]. Items are scored between 1 (definitely agree) and 4 (definitely disagree). After reverse-scoring, the higher total scores reflect the presence of more autistic traits. Additionally, in line with the original validation of the Dutch AQ [13], we calculated scores on the “social interaction” and “attention to detail” subscales. Higher scores on these subscales indicate greater difficulties in social interactions, and a greater attention to, and interests in, patterns and details, respectively. The AQ was completed by both samples.

### 2.3. MRI and PET/CT Acquisition and Processing

Details of magnetic resonance imaging (MRI) and PET/CT acquisitions and processing steps have been previously described [9]. In short, a structural T1-weighted MRI scan was obtained on a 3T Ingenia (Philips Healthcare, Best, The Netherlands). A 90 min dynamic PET scan was obtained on a Biograph Horizon with TrueV option (Siemens Healthineers, Erlangen, Germany) or Vereos (Philips Healthcare, Best, The Netherlands), directly after the administration of approximately 150 MBq [^18^F]-FDOPA. A low dose CT scan (110/120 kVp, 35 mAs) was acquired for attenuation–correction purposes. Participants consumed 150 mg of carbidopa and 400 mg entacapone, 1 hour before starting the PET/CT scan.

We used [^18^F]-FDOPA uptake in gray matter (GM) cerebellum as a reference to calculate the influx constant (*k*_i_^cer^ min^−1^; hereon labeled as *k*_i_^cer^) throughout the brain using reference Patlak graphical analysis [14]. The ROIs were automatically identified from the co-registered MRI scan using PVElab (v2.3; Neurobiology Research Unit, Copenhagen, Denmark; [15,16]), using a maximum probability atlas [17]. The ROIs included the GM of the whole striatum and three striatal anatomical sub-regions (putamen, nucleus accumbens, and caudate nucleus), which were selected on the basis of their putative role in ASD [2,3], and their reliability in terms of imaging [^18^F]-FDOPA uptake [18].

In addition to the ROI analysis, the parametric image of each participant was transformed to standard space to facilitate voxel-based comparisons. To do so, we first normalized the participant’s co-registered MRI scan using SPM12 (Institute of Neurology, London, UK). The resulting transformation matrix was applied to the parametric image, which was then smoothed using an 8 mm full width at half maximum (FWHM) Gaussian filter [18].

### 2.4. Statistical Analysis

With reference to the ROI method, data were analyzed using JASP version 0.16 [19]. We used multiple linear regression analysis to compare the regional *k*_i_^cer^ values between ASD and controls and to assess the associations of *k*_i_^cer^ with the AQ total and subscale scores. We adjusted for four confounders: age, sex, smoker status (yes/no), and scanner type (Vereos/Biograph Horizon). A two-tailed *p*-value of 0.05 was used to evaluate the statistical significance. In addition, Bayesian analyses were conducted (see Appendix A). Voxel-based comparisons were made in SPM12. An independent samples *t*-test was used to examine group differences in *k*_i_^cer^, and multiple linear regression analysis was used to examine associations between *k*_i_^cer^ and AQ scores. Confounders included age, sex, smoker status, and scanner type. A family-wise error (FWE) rate of α = 0.05 was used to evaluate the statistical significance. We conducted several additional control analyses to assess the robustness of the findings. First, we repeated our analyses restricting ourselves to voxels with *k*_i_^cer^ values above 0.001 and 0.005, effectively excluding voxels showing little specific [^18^F]-FDOPA uptake. Second, we repeated our analyses with unsmoothed data and after smoothing with a 4 mm FWHM Gaussian filter (i.e., instead of the 8 mm FWHM Gaussian filter that we used for the main analysis). Third, we conducted the analyses for the two PET/CT scanners separately.

## 3. Results

### 3.1. Sample Characteristics

Sample characteristics are reported in Table 1. Participants with ASD had significantly higher AQ total scores (*t*_64_ = 8.74, *p* < 0.001), as well as social interaction (*t*_64_ = 8.34, *p* < 0.001) and attention to detail (*t*_64_ = 5.55, *p* < 0.001) subscale scores. With regard to self-reported lifetime diagnosed mental health conditions, in the control group, participants reported having ever been diagnosed with a depressive disorder (*n* = 1) and anxiety disorder (*n* = 1). In the ASD sample, participants reported having ever been diagnosed with a depressive disorder (*n* = 9), attention deficit/hyperactivity disorder (*n* = 4), anxiety disorder (*n* = 2), and post-traumatic stress disorder (*n* = 2). None of the participants had currently or recently used any medication for these conditions.

### 3.2. ROI Analyses

Table 2 shows the *k*_i_^cer^ values in striatal ROIs and their associations with the AQ total and subscale scores. We found no significant differences in the striatal *k*_i_^cer^ values between ASD and controls. Moreover, within the control sample, and within the combined ASD and control sample, we found no significant associations between the AQ scores and striatal *k*_i_^cer^ values. In contrast, in the ASD sample, *k*_i_^cer^ values in the whole striatum, putamen, and nucleus accumbens were significantly negatively associated with AQ attention to detail subscale scores. These associations remained negative when we examined the results without adjusting for confounders or for the two PET/CT scanners separately, although they became statistically non-significant. No other statistically significant associations were observed.

Bayesian analyses supported the observed null findings over the alternative hypotheses (Appendix A). Notably, these analyses also did not provide support for a relationship between AQ attention to detail subscale scores and *k*_i_^cer^ values.

### 3.3. Voxel-Based Comparisons

Figure 1 (panels A and B) shows the average *k*_i_^cer^ for ASD and control participants in voxels throughout the brain. In striatal as well as extrastriatal regions, we found no statistically significant differences in the *k*_i_^cer^ values between ASD and controls, and in neither sample did we observe significant associations between the *k*_i_^cer^ values and the AQ total or social interaction subscale scores. These results were similar, regardless of whether we adjusted for confounders, applied varying *k*_i_^cer^ thresholds, used unsmoothed data or data smoothed with a 4 FWHM Gaussian filter, or examined the results for the two PET/CT scanners separately. We did observe, in accordance with the ROI analysis, that *k*_i_^cer^ values in a small cluster of voxels in the left nucleus accumbens, significantly negatively correlated with the AQ attention to detail subscale in the ASD sample (and not in controls) (Figure 1C). At more lenient *p*-value thresholds, this association extended to larger parts of the striatum bilaterally.

## 4. Discussion

In this exploratory study, we found no significant differences in the striatal and extrastriatal [^18^F]-FDOPA uptake between unmedicated autistic adults and controls. In the ASD sample, but not in the control or combined samples, the AQ attention to detail subscale scores were significantly and negatively correlated with dopamine synthesis capacity in the whole striatum, the putamen, and particularly the nucleus accumbens, although these findings were not supported by Bayesian analyses.

The results of our exploratory analyses confirm our previous ROI analysis, in which we found no differences in [^18^F]-FDOPA uptake in the striatum and its functional sub-regions (i.e., associative, limbic, and sensorimotor striatum) between ASD and controls [9]. We extend these findings by showing that [^18^F]-FDOPA uptake does not differ in anatomical sub-regions of the striatum (i.e., putamen, nucleus accumbens, and caudate nucleus) nor in extrastriatal brain regions. In the striatum, [^18^F]-FDOPA is decarboxylated to fluorodopamine through amino acid decarboxylase (AADC) and stored in vesicles within presynaptic terminals [20]. Although it is well-established that striatal [^18^F]-FDOPA uptake represents dopamine synthesis capacity, the radiotracer is taken up and stored by all AADC-containing, monoaminergic neurons [21]. On the one hand, this can be considered a limitation of the method, since to some extent it remains unknown what [^18^F]-FDOPA uptake in extrastriatal regions reflects. On the other hand, since we observed no significant difference in uptake in the whole brain, this can indicate that, for instance, also serotonergic functioning in the raphe nuclei is unaltered in ASD [22].

Our findings partially differ from the study by Ernst et al. [5], who reported a decreased [^18^F]-FDOPA uptake in the anterior medial prefrontal cortex in autistic children (*n* = 14), and from the study by Nieminen von Wendt et al. [6], who found an increased [^18^F]-FDOPA uptake in the striatal and frontal regions in adults with Asperger syndrome (*n* = 8). Future studies can assess whether the differences between samples in factors, such as age, ASD diagnosis, and symptom severity, can have contributed to these partially discrepant findings.

Although we found no group differences in [^18^F]-FDOPA uptake, we did find significant negative associations between AQ attention to detail subscale scores and dopamine synthesis capacity in striatal ROIs among ASD adults. These findings should be interpreted with caution since multiple tests were performed, and Bayesian analyses were inconsistent with these observations. Nevertheless, it is of interest that a recent study also showed that in individuals with ASD (*n* = 18), but not in controls (*n* = 20), striatal dopamine D_1_ receptor binding was negatively associated with the same AQ attention to detail subscale [23]. This finding can be accounted for by either increased endogenous dopamine or by the expression of fewer D_1_ receptors. This latter explanation seems more plausible, since the authors note that the assessment of D_1_ receptor binding is unlikely to be strongly influenced by the availability of endogenous dopamine. Our findings of no increased striatal dopamine synthesis capacity in ASD, and a recent report of a decreased striatal dopamine release in response to monetary reward in adults with ASD (*n* = 10) compared to controls (*n* = 12) [24], support this interpretation, as these suggest that endogenous synaptic dopamine is not higher in ASD. Together, these findings can then be interpreted as indicating that a reduction in striatal dopamine signaling is associated with attentional processes relevant to ASD, which accords with previous theoretical and empirical work on the role of striatal dopamine in ASD [1,2,3,25].

If striatal dopamine is indeed related to attentional processing in ASD, and we emphasize that this finding requires replication in an independent cohort, then it can do so in different ways. For example, striatal dopamine can be involved in the direction of attention to salient information [25,26,27], which would fit with our finding that associations were strongest in the nucleus accumbens, a region known to play a role in these cognitive processes [28]. It is also possible that our findings reflect alterations secondary to the perturbations in other neurotransmitter systems. Future studies, combining molecular imaging methods with objective assessments of cognitive functioning would be useful in this respect. Of note, there has been a relative scarcity of molecular imaging studies in ASD [4], and future (preferably longitudinal) assessments of different aspects of the dopamine and other neurotransmitter systems would help elucidate the role of these systems in ASD. Such assessments can also increase our knowledge on the reasons why certain medications might (not) work in autistic individuals. Given their high prescription rates [29], this seems useful and necessary.

The strengths of the present study are its large sample size and the completion of additional analyses to ensure the robustness of our findings. Note that, as reported previously [9], the mean cerebellar standardized uptake values were comparable for ASD and controls and, therefore, possible differences in non-specific uptake of [^18^F]-FDOPA can be excluded. A first limitation of the study is the exclusion of participants who used medication or had been diagnosed with a low IQ or a psychotic disorder, as we do not know how our findings generalize to those populations. Second, autistic traits were assessed by self-report only. Third, since the study was exploratory, we did not conduct a priori sample size calculations for the present study purposes. Fourth, data were collected on two PET/CT systems. However, reconstruction parameters for the two scanners were harmonized using published guidelines [30], and scanner type was added as a covariate to the analyses. Fifth, we chose to use [^18^F]-FDOPA analyses methods based on previous literature (e.g., [18]) and conducted additional sensitivity analyses (e.g., with varying *k*_i_^cer^ thresholds); however, future studies should explore the added value of more sophisticated analysis methods. For example, the role of partial volume correction should be further investigated, as some studies have indicated GM/WM differences between autistic individuals and healthy controls [31,32].

In conclusion, our exploratory findings indicate that [^18^F]-FDOPA uptake in the brain does not significantly differ between autistic adults and controls. The striatal dopamine synthesis capacity can be negatively associated with scores on the AQ attention to detail subscale in autistic adults, but replication of this finding is necessary.

## Figures and Tables

**Figure 1 diagnostics-11-02404-f001:**
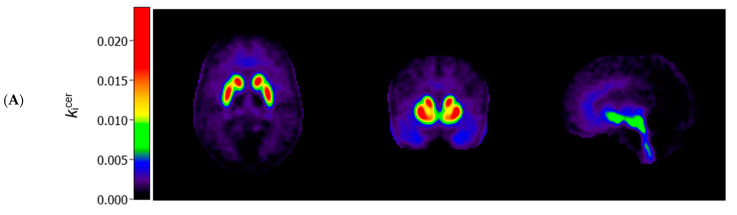
Axial (left), coronal (middle), and sagittal (right) view of the mean cerebral [^18^F]-FDOPA uptake (*k*_i_^cer^ min^−1^; unadjusted and unsmoothed) in (**A**) adults with autism spectrum disorder (*n* = 44) and (**B**) controls (*n* = 22). Panel (**C**) shows statistically significant negative associations between scores on the autism spectrum quotient attention to detail subscale and *k*_i_^cer^ values in autistic adults, overlaid on a single subject T1-weighted MRI scan, when family-wise error (FWE) rate-adjusted or unadjusted *p*-values of 0.05 are used (8 mm smoothing, threshold of *k*_i_^cer^ ≥ 0.005).

**Table 1 diagnostics-11-02404-t001:** Sample characteristics of adults with autism spectrum disorder (ASD) and controls.

Variable	ASD (*n* = 44)	Controls (*n* = 22)
Male, no. (%)	28 (64%)	14 (64%)
Age in years, mean (SD)	23.74 (2.64)	23.47 (2.48)
IQ, mean (SD)	103.75 (5.19)	105.05 (4.90)
Smoker, no. (%)	2 (5%)	1 (5%)
Scanned on Vereos PET/CT scanner, no. (%)	31 (70%)	13 (59%)
Approximate injected [^18^F]-FDOPA dose in MBq, mean (SD)	161.55 (7.26)	157.24 (8.57)
AQ total score, mean (SD)	132.41 (20.05)	91.73 (12.01)
AQ social interaction subscale, mean (SD)	105.25 (17.65)	71.27 (10.22)
AQ attention to detail subscale, mean (SD)	27.16 (4.94)	20.45 (4.19)

SD, standard deviation; IQ, intelligence quotient; MBq, megabecquerel; and AQ, autism spectrum quotient.

**Table 2 diagnostics-11-02404-t002:** Striatal [^18^F]-FDOPA uptake (*k*_i_^cer^ min^−1^) in ASD adults and controls, and its association with self-reported autistic traits.

				Association Between *k*_i_^cer^ Value and AQ Scores								
	*k*_i_^cer^, Mean (SD)			ASD			Controls			Combined Sample		
ROI	ASD	Controls	*p*-value	Total	Social	Detail	Total	Social	Detail	Total	Social	Detail
Whole striatum	0.0145 (0.0023) ^a^	0.0143 (0.0024) ^a^	0.87 ^a^	*β* = −0.04, *p* = 0.81	*β* = 0.04, *p* = 0.80	*β* = −0.35, *p* = 0.04	*β* = 0.08, *p* = 0.74	*β* = 0.08, *p* = 0.74	*β* = 0.03, *p* = 0.90	*β* = 0.02, *p* = 0.87	*β* = 0.06, *p* = 0.65	*β* = −0.14, *p* = 0.28
Putamen	0.0157 (0.0025)	0.0153 (0.0026)	0.61	*β* = 0.03, *p* = 0.86	*β* = 0.12, *p* = 0.46	*β* = −0.36, *p* = 0.04	*β* = 0.10, *p* = 0.67	*β* = 0.06, *p* = 0.80	*β* = 0.13, *p* = 0.56	*β* = 0.10, *p* = 0.43	*β* = 0.14, *p* = 0.28	*β* = −0.08, *p* = 0.52
Nucleus accumbens	0.0114 (0.0024)	0.0108 (0.0024)	0.38	*β* = −0.09, *p* = 0.58	*β* = 0.00, *p* = 0.99	*β* = −0.43, *p* = 0.01	*β* = 0.13, *p* = 0.59	*β* = 0.14, *p* = 0.57	*β* = 0.04, *p* = 0.85	*β* = 0.09, *p* = 0.49	*β* = 0.12, *p* = 0.33	*β* = −0.10, *p* = 0.45
Caudate nucleus	0.0135 (0.0022)	0.0137 (0.0025)	0.69	*β* = −0.12, *p* = 0.48	*β* = −0.06, *p* = 0.72	*β* = −0.29, *p* = 0.09	*β* = 0.05, *p* = 0.84	*β* = 0.10, *p* = 0.69	*β* = −0.09, *p* = 0.72	*β* = −0.09, *p* = 0.48	*β* = −0.05, *p* = 0.66	*β* = −0.19, *p* = 0.12

ASD, autism spectrum disorder; AQ, autism spectrum quotient; SD, standard deviation; ROI, region of interest; total, AQ total scores; social, AQ social interaction subscale scores; and detail, AQ attention to detail subscale scores. Analyses adjusted for age, sex, smoking status, and PET/CT scanner type. ^a^, as reported in [9].

## Data Availability

The datasets generated during and/or analyzed during the current study are available from the corresponding author upon reasonable request.

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
