# Peer review of "Cerebral [18F]-FDOPA Uptake in Autism Spectrum Disorder and Its Association with Autistic Traits"

_diagnostics, 2021, doi:10.3390/diagnostics11122404_

Round 1
Reviewer 1 Report
Brain AADC activity was estimated by [18F]-FDOPA PET in 44 young adults with autism spectrum disorder (ASD) and 22 controls. Region-of-interest and voxel-based analyses showed no statistically significant differences in Ki between groups. In ASD, striatal Ki was negatively associated with AQ attention.
This PET study is based on a reasonable number of observations, larger than samples currently reported in the literature. It provides an interesting contribution to the field, although in its present form, the results do not fully support the authors’ conclusions.
The manuscript would be much improved if the authors could provide an adequate reply to the following suggestions:
- It is not known whether ASD patients were medically treated. In this case, this should be mentioned and taken into account in the analyses.
- The authors did not correct for partial volume effect. Although the latter is unlikely for dorsal striatum, it can alter the estimation of NAc KI. The authors should provide analyses with and without correction for PVE.
- As for multiple linear regressions (on ROIs and voxelwise), the authors should systematically report:
- In an analysis gathering all observations:
- Simple effects of group
- Group x scanner interaction . Indeed, there is an uneven proportion of patients across scanners. A significant interaction would lead to reservations concerning the main effects.
- In an analysis focused on patients:
- The effects of AQ scores
- The absence of significant difference between groups does not prove the absence of group difference. The authors should provide Bayesian inference of the probability of difference between groups in the relevant regions.
- In an analysis gathering all observations:
- The authors should consider running a VBM analysis on their MR data. Increased bilateral temporal and right dorsolateral prefrontal grey matter volume was reported in ASD. These areas might be used as priors for Ki statistical inference.
- The reviewer does not understand the rationale for conducting repeated analyses with unsmoothed data and after smoothing with a 4 mm FWHM Gaussian filter. This looks like fishing expeditions.
Author Response
Response to Reviewer 1 comments
Brain AADC activity was estimated by [18F]-FDOPA PET in 44 young adults with autism spectrum disorder (ASD) and 22 controls. Region-of-interest and voxel-based analyses showed no statistically significant differences in Ki between groups. In ASD, striatal Ki was negatively associated with AQ attention.
This PET study is based on a reasonable number of observations, larger than samples currently reported in the literature. It provides an interesting contribution to the field, although in its present form, the results do not fully support the authors’ conclusions.
The manuscript would be much improved if the authors could provide an adequate reply to the following suggestions:
- It is not known whether ASD patients were medically treated. In this case, this should be mentioned and taken into account in the analyses.
Author reply
We thank the reviewer for this comment. We can confirm that neither ASD patients nor controls had currently or recently used medication. This is now briefly mentioned in the Methods (section 2.1, p. 4), Results (section 3.1, p. 7), Discussion (first paragraph, p. 9), and Supplementary Methods 1.
- The authors did not correct for partial volume effect. Although the latter is unlikely for dorsal striatum, it can alter the estimation of NAc KI. The authors should provide analyses with and without correction for PVE.
Author reply:
We thank the Reviewer for this suggestion. We agree with the Reviewer that PVE correction could be of interest, as it could potentially increase the ki values (i.e., improve the quantification) of smaller brain structures such as the nucleus accumbens. Nevertheless, we had several reasons not to perform PVE correction in the current study. First, in order to achieve a good repeatability and be consistent with previous literature, analyses were conducted based on previous validation studies of [18F]-FDOPA data processing, in which no PVE correction was conducted (e.g., Egerton et al., 2010, Neuroimage). Second, striatal volumes of interest (i.e., whole striatum and anatomical subregions) were of comparable size in autistic (e.g., bilateral whole striatum M = 15.0 mL, SD = 1.2) and non-autistic participants (M = 15.6 mL, SD = 1.6), and thus we expect that PVE correction would have not greatly influenced their relative association. Third, PVE correction can reduce signal-to-noise ratios and introduce Gibbs artefacts (Rahmin et al., 2013, Med Phys), which could potentially hamper the reproducible and accurate estimation of ki. Fourth, at this moment, unfortunately the Siemens Biograph Horizon PET-system is no longer in use at our hospital, and so reconstruction using PVE correction can no longer be completed for this dataset using resolution recovery-based methods. PVE correction should then be performed on (already processed and Gaussian-smoothed) data using e.g., Van Cittert deconvolution, potentially reducing signal-to-noise. As such, given the limitations surrounding current PVE-corrected analyses, we opted to not conduct further PVE correction, but we agree with the author that such analyses might be interested for future work, in particular in more technical papers comparing the effects of doing so. Therefore, we have added the absence of PVE correction in this paper as a potential limitation in the Discussion section (p. 11).
- As for multiple linear regressions (on ROIs and voxelwise), the authors should systematically report:
- In an analysis gathering all observations:
- Simple effects of group
- Group x scanner interaction . Indeed, there is an uneven proportion of patients across scanners. A significant interaction would lead to reservations concerning the main effects.
- In an analysis focused on patients:
- The effects of AQ scores
Author reply
We thank the reviewer for their comments and believe that we now have most of these analyses in the paper. Simple effects of group and association between dopamine synthesis capacity and AQ scores in ASD and control samples have been added to Table 1. With regard to Group x Scanner interaction, we would like to note that we harmonized PET reconstruction procedures for both scanners using previously published guidelines (Verwer et al., 2021, Eur J Nucl Med Mol Imaging) that are now being transformed into a recently introduced EARL accreditation program for 18F/11C Brain PET studies (https://earl.eanm.org/18f-brain-pet-ct/), and that we adjusted for scanner-type in the analyses. We have additionally checked the results and can confirm that they are of similar magnitude on both scanners. We did find that the significant association between striatal kicer and AQ attention to detail subscale scores in the ASD sample was non-significant when assessed on the separate scanners, but associations remained present in the same negative direction (Siemens: β = -0.15; Philips: β = -0.28). In sum, conducting analyses separately for the Siemens and Philips PET-scanner were comparable with the combined analyses reported in the paper. We now briefly describe this in the Discussion (p. 11).
- The absence of significant difference between groups does not prove the absence of group difference. The authors should provide Bayesian inference of the probability of difference between groups in the relevant regions.
Author reply
We thank the Reviewer for this suggestion and have added the results of Bayesian analyses as a Supplementary Table to the manuscript. These analyses largely confirm our previous results, although evidence for the observed statistically significant finding of an association between AQ attention to detail scores and striatal dopamine synthesis capacity appears to be lacking. We have described these methods and results (and their implications) throughout the manuscript.
- The authors should consider running a VBM analysis on their MR data. Increased bilateral temporal and right dorsolateral prefrontal grey matter volume was reported in ASD. These areas might be used as priors for Ki statistical inference.
Author reply:
We thank the author for this suggestion. We have conducted a preliminary analysis on GM and WM differences between patients and controls in multiple regions of interest (automatically extracted from the Hammers template), and this did not show significant group differences. As such, we do not expect that further taking into account GM-volumes will substantially influence our results. However, to be certain of this, we agree with the Reviewer that further analyses could be conducted, in particular in a more technical investigation of the role of volumetry in PET analyses. We now acknowledge this in the Discussion (p. 11).
- The reviewer does not understand the rationale for conducting repeated analyses with unsmoothed data and after smoothing with a 4 mm FWHM Gaussian filter. This looks like fishing expeditions.
Author reply
We understand the Reviewer’s concern about the multitude of analyses we conducted on the same data. The reason for including additional analyses such as those with alternative smoothing parameters, is that various approaches to [18F]-FDOPA PET processing are available (this is acknowledged and explored e.g. in Egerton et al., 2010, Neuroimage; Avram et al., 2019, Brain; Veronese et al., 2021, Neuropsychopharmacol). Therefore, we repeated the analyses with different parameters to assess whether our findings were robust to different ways of data processing. We have added an additional comment in the Methods section to clarify this (p. 6).

Reviewer 2 Report
The article is very interesting and well written. I believe that more information about control group should be provided
Reviewer 3 Report
The communication entitled “Cerebral [18F]-FDOPA Uptake in Autism Spectrum Disorder and its Association with Autistic Traits”, reported an extension of a previous study using [18F]-FDOPA PET/CT to examine cerebral [18F]- 17 FDOPA influx in exploratory region-of-interest including striatal and extrastriatal regions in 44 adults diagnosed with autism spectrum disorder (ASD) and 22 controls, aged 18 to 30 years in order to examine their associations with self reported autistic traits, assessed by a self-report questionnaire, the Autism Spectrum Quotient (AQ). The authors found no significant differences in striatal and extrastriatal [18F]-FDOPA uptake between ASD adults and controls and in the autistic group found that AQ attention to detail subscale scores were significantly and negatively correlated with dopamine synthesis capacity in the whole striatum, the putamen, and particularly the nucleus accumbens,
There are several limitations that the authors mentioned as the use of radiotracer stored by all AADC-containing monoaminergic neurons, the inclusion of autistic patients without psychotic symptoms, the autistic traits assessed by self-report questionnaire only, the lack of cognitive assessments. Therefore, the hypotesis suggested by the authors remains a hypothesis that must will be proved.
It is not clear if the ASD patients were under treatment or presenting with other psychiatric comorbidities. It is important to add these informations that they were missing in the inclusion and exclusion criteria. Why were not considered in the ASD group also ADOS II scores? .
However the topic is of interest. I suggest to convert this communication in a letter to editor.
Author Response
Response to Reviewer 3 comments
The communication entitled “Cerebral [18F]-FDOPA Uptake in Autism Spectrum Disorder and its Association with Autistic Traits”, reported an extension of a previous study using [18F]-FDOPA PET/CT to examine cerebral [18F]- 17 FDOPA influx in exploratory region-of-interest including striatal and extrastriatal regions in 44 adults diagnosed with autism spectrum disorder (ASD) and 22 controls, aged 18 to 30 years in order to examine their associations with self reported autistic traits, assessed by a self-report questionnaire, the Autism Spectrum Quotient (AQ). The authors found no significant differences in striatal and extrastriatal [18F]-FDOPA uptake between ASD adults and controls and in the autistic group found that AQ attention to detail subscale scores were significantly and negatively correlated with dopamine synthesis capacity in the whole striatum, the putamen, and particularly the nucleus accumbens,
- There are several limitations that the authors mentioned as the use of radiotracer stored by all AADC-containing monoaminergic neurons, the inclusion of autistic patients without psychotic symptoms, the autistic traits assessed by self-report questionnaire only, the lack of cognitive assessments. Therefore, the hypotesis suggested by the authors remains a hypothesis that must will be proved.
Author reply:
We thank the reviewer for these comments and agree that, as mentioned in the Discussion, the results in extra-striatal regions need to interpreted with some caution, that we tested a subset of the ASD population, and that no cognitive assessments were made. We hope and expect that these findings will lead to subsequent follow-up studies of dopamine function in ASD, to further explore these matters.
- It is not clear if the ASD patients were under treatment or presenting with other psychiatric comorbidities. It is important to add these informations that they were missing in the inclusion and exclusion criteria.
Author reply:
We thank the Reviewer for this suggestion and have added further information on co-occurring mental health conditions to the paper (Results, 3.1 Sample characteristics, p. 7).
- Why were not considered in the ASD group also ADOS II scores? .
Author reply:
Indeed, it would be possible to conduct additional analyses with ADOS-II scores. However, for various reasons we decided not to include those analyses here:
- Given the wide range of available AQ scores, and multiple regions of interest tested, a large number of tests were already planned, and inclusion of analyses with ADOS-II scores would have further increased the possibility of observing false positive results.
- ADOS-II scores were only available for the ASD sample, and as such, those analyses could not be conducted within the control group.
- Module 4 (for adults) ADOS-II calibrated severity scores remain relatively poorly studied. Although this module is recommended as a gold standard instrument for ASD diagnosis in adults, validation of its subscale calibrated severity scores is somewhat lacking compared to other modules (e.g., Pugliese et al., 2015; J Autism Dev Disord). This is particularly relevant when considering the scoring of the ‘Restricted and Repetitive Behaviors’ subscale, which relies on the observation of relatively rare behaviors in a short time frame. Therefore, the ADOS-II might not be the optimal instrument to assess these behaviors in adults (e.g., Hus & Lord, 2014, J Autism Dev Disord).
- However the topic is of interest. I suggest to convert this communication in a letter to editor.
Author reply:
We thank the Reviewer for these further comments. We would be happy to convert the paper into a ‘Letter to the Editor’, but as far as we can tell, the journal Diagnostics does not offer this option in the submission process. Rather, they state that they use the ‘Communication’ format for ‘preliminary, but significant, results’ (https://www.mdpi.com/journal/diagnostics/instructions), which out of all available article types we think best describes the present paper.
Reviewer 4 Report
The presented article includes a correct introduction, an appropriate methodology, with an experimental group and a control group, a pertinent data analysis, and appropriate tables and graphs.
For all this I recommend its publication.
Round 2
Reviewer 3 Report
The authors have addressed all my points